# Heavily Armed Ancestors: CRISPR Immunity and Applications in Archaea with a Comparative Analysis of CRISPR Types in Sulfolobales

**DOI:** 10.3390/biom10111523

**Published:** 2020-11-06

**Authors:** Isabelle Anna Zink, Erika Wimmer, Christa Schleper

**Affiliations:** Department of Functional and Evolutionary Ecology, Archaea Biology and Ecogenomics Unit, University of Vienna, 1090 Vienna, Austria; erika.wimmer@univie.ac.at (E.W.); christa.schleper@univie.ac.at (C.S.)

**Keywords:** CRISPR, archaea, type III, sulfolobales, cOA-signaling, viruses, CRISPR model organisms, CRISPR applications

## Abstract

Prokaryotes are constantly coping with attacks by viruses in their natural environments and therefore have evolved an impressive array of defense systems. Clustered Regularly Interspaced Short Palindromic Repeats (CRISPR) is an adaptive immune system found in the majority of archaea and about half of bacteria which stores pieces of infecting viral DNA as spacers in genomic CRISPR arrays to reuse them for specific virus destruction upon a second wave of infection. In detail, small CRISPR RNAs (crRNAs) are transcribed from CRISPR arrays and incorporated into type-specific CRISPR effector complexes which further degrade foreign nucleic acids complementary to the crRNA. This review gives an overview of CRISPR immunity to newcomers in the field and an update on CRISPR literature in archaea by comparing the functional mechanisms and abundances of the diverse CRISPR types. A bigger fraction is dedicated to the versatile and prevalent CRISPR type III systems, as tremendous progress has been made recently using archaeal models in discerning the controlled molecular mechanisms of their unique tripartite mode of action including RNA interference, DNA interference and the unique cyclic-oligoadenylate signaling that induces promiscuous RNA shredding by CARF-domain ribonucleases. The second half of the review spotlights CRISPR in archaea outlining seminal in vivo and in vitro studies in model organisms of the euryarchaeal and crenarchaeal phyla, including the application of CRISPR-Cas for genome editing and gene silencing. In the last section, a special focus is laid on members of the crenarchaeal hyperthermophilic order Sulfolobales by presenting a thorough comparative analysis about the distribution and abundance of CRISPR-Cas systems, including arrays and spacers as well as CRISPR-accessory proteins in all 53 genomes available to date. Interestingly, we find that CRISPR type III and the DNA-degrading CRISPR type I complexes co-exist in more than two thirds of these genomes. Furthermore, we identified ring nuclease candidates in all but two genomes and found that they generally co-exist with the above-mentioned CARF domain ribonucleases Csx1/Csm6. These observations, together with published literature allowed us to draft a working model of how CRISPR-Cas systems and accessory proteins cross talk to establish native CRISPR anti-virus immunity in a Sulfolobales cell.

## 1. Introduction—Where There Is a Cell, There Is a Virus

From the biggest animals to the smallest microbe, meaning from the complex multicellular to simpler unicellular life forms, viruses are found as infective agents. Intriguingly, the particles of prokaryote-infecting viruses (i.e., those infecting archaea and bacteria) outnumber their hosts by at least an order of magnitude in aquatic systems [1]. Considering that bacteria and archaea constitute the majority of all living cells, viruses should be a predominant biological entity on Earth [2]. Viruses, plasmids, and transposons can be designated as mobile genetic elements (MGE) and consist primarily of nucleic acids which they need to replicate and express inside a living host. This dependence on a cellular replication machinery and the absence of a bona fide cell structure precludes them from being considered as living organisms. In the course of evolution, viruses and their host cells (both eukaryotic and prokaryotic) have shaped each other due to lateral gene transfers, recombination events and co-evolution. While cellular genomes often harbor viral traces, cellular genes are rather rarely found in viral genomes [3] although they sometimes transmit novel physiological traits [4]. In mammals, up to 50% of genome content stems from exogenous origin [5], also including “real” viral genes. Several bacteria gained their pathogenicity against eukaryotic cells through uptake of viral genes. Viruses can spread within and kill the most dominant populations and thus can contribute to maintaining microbial diversity in aquatic environments (‘Killing the winner’) [6]. Many bacteria even release viral toxins or intact viruses against other, sensitive bacterial competitors to invade new territories or dominate the habitat [7,8]. This shows that viruses play an immense role in prokaryotic populations and therefore greatly shape the microbial communities. They “seek and destroy” which drives the selective pressure on microbes towards strategies to withstand viral threats, i.e., becoming immune. Prokaryotes have evolved several ways to protect themselves from viral or plasmid infection, such as the well-known restriction–modification (R-M) system (reviewed in [9]), virus entry inhibitors (reviewed in [10]), “abortive phage infection” (via toxin/anti-toxin systems) (reviewed in [11]), Argonautes that destruct MGEs via DNA-mediated DNA interference [12], and other defense system, such as bacteriophage exclusion (BREX) or defense island system associated with restriction-modification (DISARM), blocking phage infection or replication by a yet unknown mechanism [13,14,15]. These systems are considered as “innate” defense, in the sense that they can act without the need for prior exposure to the virus [13,14]. As in many of those systems, interference is dependent on the recognition of a predetermined sequence pattern [9,13,14], they are often inefficient or overcome by the co-evolving virus [16]. One may ask, why very closely related strains sharing an almost identical genome often show differences in virus/plasmid susceptibility, or why strains sensitive to a certain virus can rapidly become resistant? Answers to these questions may lie in another immune system taking prokaryotic immunity to the next level: The CRISPR system. In contrast to innate immune defense, the Clustered Regularly Interspaced Short Palindromic Repeats (CRISPR) system is adaptive meaning that it allows the prokaryotes to generate an immunological memory of infections, reminiscent of immune systems in eukaryotic organisms. Viral encounters are stored as short DNA pieces in the prokaryote′s genome, rendering the cell immune for a second round of viral infection. Intriguingly, this system may, to a certain extent, fit a Lamarckian view of evolution, since a single cell′s genome is directly modified, and the same cell is affected by the modification in a beneficial way [17]. Moreover, the immunological record is inherited to the progeny of that cell imparting the resistance against the virus [17].

## 2. CRISPR Immunity in Prokaryotes—Archaea and Bacteria in the Ring with Viruses

CRISPR-Cas systems occur in the genomes of ~40% of bacteria and ~80% of archaea [18]. They are very diverse among species [19], but all consist of structurally conserved CRISPR arrays and *Cas* (CRISPR-associated) genes encoding CRISPR proteins driving the immune response. *Cas* genes and their proteins were used to divide CRISPR systems into two distinct classes, six types and 33 subtypes [19], with class I representing the most abundant prokaryotic CRISPR-Cas systems, found in almost all CRISPR-equipped archaea and many bacteria (cf. Figure 1). CRISPR class II, which also accommodates the hallmark genome editing tool Cas9, only accounts for ~15% of naturally found CRISPR-Cas systems. Almost all of those fall into the bacterial domain (Figure 1) [18]. For detailed insights into CRISPR-Cas classification and relative distribution among prokaryotic genomes, we refer the reader to the recent review by Makarova and colleagues, which we used as classification signpost throughout this review [19]. CRISPR array architecture is conserved among the different types and starts with an AT-rich leader sequence exhibiting promoter function [20,21,22], followed by a progression of array-specific repeat sequences interspaced by mostly unique 30–45 bp DNA sequences called “spacers” (Figure 1). Among all prokaryotic spacers sequenced, only 7% could be matched to public databases (even when mismatches were allowed [23]) and those were shown to predominantly originate from MGEs, such as viruses or plasmids [24]. This finding clearly suggest that CRISPR primarily serves as a defense system against MGEs and further illustrates the overwhelmingly huge “dark matter” of unknown MGEs stored in the vast majority of spacers [23,24].

CRISPR immunity can be divided into three main steps, known as “Adaptation”, “Processing” and “Interference” (Figure 1). In the adaptation step, the infecting virus is recognized as intruder and a short piece of its DNA is incorporated as spacer into the CRISPR array in order to establish an adaptive immunological memory of the “enemy”. Next, the spacer is transcribed and further cleaved into a CRISPR RNA (crRNA) in the “Processing” step (Figure 1). The mature crRNA incorporates into CRISPR effector proteins, generating virus-specific “weapons” that constitutively patrol the cell to be ready for a counterstrike. The “Interference” step is initiated upon a second infection of a cognate virus DNA or RNA which is recognized and bound by a crRNA and further degraded by the catalytic action of the assembled effector proteins (Figure 1). Notably, besides target cleavage, some CRISPR effector proteins can be activated to induce promiscuous cleavage of host and invader nucleic acids, known as collateral damage [30]. The individual steps of CRISPR immunity with a focus on interference are discussed below.

### 2.1. CRISPR Adaptation—Know Your Enemy

Acquiring spacers from a freshly infecting virus represents the first crucial step defining CRISPR adaptive immunity. De novo spacer acquisition was observed predominantly within regions of free DNA ends provided on the linearized virus DNA upon injection, or upon formation of double strand breaks [31,32,33]. From these regions, the CRISPR adaptation complex specifically recognizes and excises a pre-spacer sequence which, in most cases, is specifically selected by the presence of a subtype (or sometimes even species)-specific recognition motif termed PAM (protospacer adjacent motif). As the PAM is a prerequisite for recognizing and discriminating between foreign and host double strand DNA (dsDNA) later in the CRISPR interference step (see below), PAM-oriented acquisition ensures selection of functional spacers only [34,35]. Following further processing, the novel spacer is integrated next to the leader sequence into the existing CRISPR array. Such polarized spacer acquisition allows to establish a chronological memory of infections, with the first spacer generally originating from the most recent invader (Figure 1). Mechanistically, the integration of novel spacers involves two cleavage-ligation reactions at the leader and spacer end of the first repeat (reviewed in [36]), executed by the heterohexameric CRISPR adaptation complex consisting of the Cas1 integrase and Cas2 nuclease [37,38] (Figure 1). *Cas1* constitutes the most highly conserved *cas* gene associated with the majority of CRISPR types of both classes and therefore serves as marker to identify active CRISPR arrays in prokaryotic genomes [19,39]. Apart from Cas1–Cas2, de novo spacer acquisition is aided by other accessory proteins in some organisms, such as non-CRISPR related host factors [40,41,42], CRISPR interference enzymes such as the bacterial Cas9 [43] or endonucleases of the Cas4 family [42,44,45,46,47,48,49]. Recently, the latter has gained increased attention in the archaeal CRISPR field, as Cas4 was found to play significant roles in PAM recognition and length determination of pre-spacers during the adaptation process in some archaeal models [42,45,47,48]. In addition, Cas4 overexpression inhibited spacer acquisition in *Saccharolobus* (formerly: *Sulfolobus*) *islandicus* [50], suggesting its potential to be exploited by viruses as an anti-CRISPR factor to inhibit CRISPR immunity [47].

Besides de novo spacer acquisition, a pre-existing spacer in the CRISPR array that fully or partially matches an incoming virus during the CRISPR type I interference reaction, can “prime” acquisition leading to a boosted immune response, as more spacers against the same intruder are collected (reviewed in [36,51]). Interference-driven acquisition is prevalent in subtype I-F systems, where Cas2 and the interference-nuclease Cas3 (see below) are fused [52]. Most interestingly, there is also evidence that mRNA of DNA viruses and the genomes of RNA viruses could be sampled for spacers, which are further reverse transcribed by a Cas1- reverse transcriptase (RT) fusion enzyme (sometimes harboring an additional Cas6 domain), and are subsequently integrated as DNA spacers into the CRISPR array [28,53,54,55]. Even if such fusion proteins have only been detected in some bacteria so far [19], bona fide RTs might as well work in concert with Cas1 in order to establish immunity against virus RNA. Indeed, CRISPR-associated RTs are located proximal to type III effector complexes capable of RNA degradation (see below) in representatives of the methanogenic archaeal genus *Methanosarcina* as well as in *Methanomethylovorans hollandica* (according to own observations based on [19]).

### 2.2. Processing of crRNAs—Forging Weapons

Upon transcription of a CRISPR array from the leader, a long precursor crRNA (CRISPR RNA) is formed and subsequently truncated within repeats into small effector crRNAs. In class I systems, Cas6 ribonucleases [56,57,58,59,60,61], Cas5d [25,62] or a host RNase E [26] are involved in crRNA maturation and cleave the repeats within specific regions that are often found structured into stem loops [63,64] (cf. Figure 1). In class 2 systems however, the interference effector proteins (see below) are involved in the maturation step and either execute this reaction intrinsically [30,65,66], or in combination with a trans-activating CRISPR RNA (tracrRNA) and host RNase III, as in Cas9 systems [67] (cf. Figure 1). In the latter, the tracrRNAs hybridizes to the partially complementary repeat sequences on the pre-crRNA, thereby licensing its cleavage solely within repeat regions by the dsRNA-specific ribonuclease RNase III [67]. Different from class II systems which can vary in length of the flanking repeat residues [64], the generic class I mature crRNA carries a defined 5′ handle derived from the preceding repeat and a 3′ residue of the downstream repeat bordering the spacer (Figure 1).

### 2.3. CRISPR Interference-Counterstrike

In the CRISPR interference step, ribonucleoprotein complexes assemble with effector Cas proteins and are guided by the crRNA to complementary DNA or RNA sequences (i.e., protospacer) of a virus invader upon a “second” infection wave (Figure 1). Through specific base pairing between crRNA and the virus protospacer, the interference machinery cleaves and eliminates the target. The interference process, target recognition and target specificity depend on the effector proteins of the specific CRISPR types, with CRISPR type I, II, V and potentially also the still ambiguous type IV primarily recognizing virus DNA substrates, while both, type III and VI identify virus RNA (Figure 1 and Figure 2). Based on different genomic and phylogenetic criteria [19], the individual types are further divided into subtypes which often vary in the their gene and catalytic domain architectures [19,68]. For instance, while the signature gene *cas3* is represented as a single gene in most type I subtypes, it is fused to *cas2* in subtype I-F (see above) or its catalytic domains are partitioned into two genes (Cas3′ and Cas3″) in subtype I-A [19]. Furthermore in subtype I-D, the type I-specific large subunit *cas8* is replaced by a type III-like large subunit *cas10d* fused to a Cas3″domain, while the other Cas3′ domain is encoded by a separate gene, therefore representing a chimera between type I and type III systems ([19,27]). Excitingly, a recent study revealed that *cas10d* internally codes for another, small subunit protein (termed Cas11d) that is required for DNA binding of the interference complex [27,69]. These extra codons were also found in *cas8* genes of type I-B and type I-C systems, indicating that internal translation from large subunits might be a widespread feature among certain type I subtypes [69]. 

In rare cases, subtypes even differ in their targeting specificities, as for instance, subtype V-G is so far the only experimentally verified type V representative found to specifically recognize RNA instead of DNA (see below and ref. [29]). Similarly, type I-D is so far the only known type I member exhibiting a dual, dsDNA and ssDNA targeting mode (see below and ref. [27]). However, as many subtypes of the different CRISPR types have not been experimentally investigated so far, such differences in target specificities remain to be examined more closely. 

#### 2.3.1. CRISPR DNA Interference 

For CRISPR systems that specifically cleave dsDNA, primary target recognition relies on the presence of a specific 2–6 bp PAM flanking the matching protospacer (reviewed in [70]). Once a cognate PAM is sensed during interrogation of the DNA by the effector proteins, the DNA is unwound, allowing for subsequent hybridization of the loaded crRNA to the protospacer sequence which subsequently licenses target cleavage [71]. Notably, PAM dependent-interference protects the cells from self-targeting because CRISPR arrays, naturally providing a match to every spacer expressed, are devoid of a functional PAM in flanking repeat sequences [72]. CRISPR type I systems in general employ a multi—subunit effector complex termed CRISPR-associated complex for antiviral defense (Cascade) which, upon PAM recognition and crRNA hybridization catalyzes protospacer degradation by the action of the Cas3 enzyme endowed with ssDNA-endonuclease (provided by an HD domain) and helicase activity. Cas3 nicks the strand opposite to the crRNA-protospacer heteroduplex and is then activated to processively cleave the upstream region [73,74,75,76,77,78,79,80,81] (c.f. Figure 1).

Type I immunity was among the first studied CRISPR interference reactions [78] with early structural and mechanistical insights for both, bacterial and archaeal Cascades [75,82]. Intriguingly, only recently, the aforementioned type I-D Cascade complex of the archaeaon *Saccharolobus islandicus* LAL 14/1 was biochemically characterized which revealed a unique chimeric cleavage mode, resembling type I and type III-targeting (see below) [27]. Besides type I–bona fide PAM-dependent cleavage of dsDNA executed by the action of a Cas3′ helicase and Cas10d endonuclease (see above)*,* the I-D Cascade backbone specifically degraded ssDNA protospacers in a ruler-like and PAM independent manner.

Type II systems act via a dual RNA guide composed of crRNA and tracrRNA together with the single-component effector Cas9 against DNA, cleaving both strands of a PAM-flanked protospacer mediated by the opposing RuvC–like and a HNH-nuclease domains [83] (cf. Figure 1). Similarly, the type V specific signature protein Cas12 carries a RuvC domain executing dsDNA cleavage of the protospacer in dependence of a PAM (cf. Figure 1). Different to type II systems however, the majority of the various type V subtypes do not require a tracrRNA [29,84] and some were shown to cleave protospacers placed on M13 phage-ssDNA in a PAM-independent manner [85] (cf. Figure 1). Remarkably, once activated by hybridization to a protospacer on ds or ssDNA, some Type V subtypes, including the well-studied effectors Cas12a and Cas12b1, executed collateral damage by non-specifically shredding ssDNA [29] (c.f. Figure 1). Such collateral damage might represent a robust way to quickly eradicate the infecting virus following the tabula rasa principle. However, indiscriminate nucleic acid cleavage also harms the host and might induce dormancy or even suicide of the infected cell. Thus, reminiscent of abortive infection, collateral damage represents an altruistic facet of the CRISPR immunity, sacrificing the infected cell for the sake of the population [86]. 

In comparison to the above-mentioned systems, the function and role of the CRISPR type IV systems is still enigmatic, as these systems are predominantly carried with bacterial plasmids and lack a nuclease domain [19]. Instead, they often encode a helicase, which was recently shown to be involved in conferring immunity against plasmids in interference assays [87]. In line with this, spacers of adjacent CRISPR arrays were found to predominantly match other plasmids, suggesting that the type IV system might specifically serve to interfere with competing plasmids [88].

60% of both, CRISPR—equipped archaea and bacteria, destroy virus DNA using Type I Cascade [19], including representatives of famous archaeal and bacterial (CRISPR) model organisms such as the archaeon *Haloferax volcanii* and *Escherichia coli*. Thus, type I systems are the most prevalent CRISPR-Cas systems in prokaryotes, whereas type II and V systems are generally underrepresented and rare in archaea. The absence of class 2 systems (including type II, V, VI) in archaea might go along with the absence of the bacteria-specific RNase III ribonuclease which is required for crRNA maturations in most of these systems (see above). Only recently, type V-F CRISPR-Cas systems were detected in uncultivated archaea of the DPANN superphylum [89] and type II-Cas9 systems were found in two representatives of the nanoarchaeota [90]. However, in case of the latter, the activity remains to be investigated in the native host, as neither in vitro nor in vivo cleavage was detected when purified from or examined in heterologous models [90].

#### 2.3.2. “RNAttack”

Different from the above-mentioned types, all subtypes of CRISPR type III and type VI systems recognize the invader′s RNA. Class II-specific type VI CRISPR-Cas systems are exclusively found in bacteria and are absent from archaea [19]. They consist of the single-component effector protein Cas13, endowed with RNase activity provided by two conserved higher eukaryotes and prokaryotes, nucleotide binding (HEPN) domains, and confer immunity against a ssRNA phage carrying a cognate protospacer [30,91,92,93]. For some, but not all subtypes, target recognition seemed to be dependent on a protospacer flanking site (PFS) located either downstream or upstream of the protospacer [30,91,94]. Initial target recognition was reported to be highly specific, as already two mismatches in the central region of the crRNA, abolished target identification [30,94,95]. This sensitivity has encouraged scientists to exploit Cas13 as tool for tracking of pathogens such as Zika virus [96], and it might be even applicable for quick detection of Covid-19 RNA [97]. Similarly to the collateral damage against ssDNA observed in type V enzymes, once activated by binding to the protospacer on the RNA, Cas13 cleaves ssRNA non-specifically [30] (cf. Figure 1). This seems to be initiated by the dramatic conformational change of the protein upon target binding, which approximates the two HEPN domains and exposes them to the environment, generating a new catalytic site indiscriminately cleaving any proximate RNA substrate [95,98]. Thus, upon specific detection of probably even lowly abundant numbers of ssRNA phages or virus transcripts, type VI effectors transform into promiscuous RNases that eradicate the virus but cause fitness loss of the cell [30,99]. A similar immune response was recently reported for a newly characterized type V effector Cas12g, extracted from a hot spring metagenome. When heterologously expressed in *E. coli*, Cas12g collaterally cleaved both, ssRNA and ssDNA following specific recognition of a protospacer on virus ssRNA [29].

Class I–specific type III systems form multiprotein complexes, which are structurally similar to the DNA-targeting type I complex Cascade, suggesting a common multi-subunit ancestor (Figure 1). Type III systems are often found in archaea (~35%) but are also common in bacteria (~25%), and therefore constitute the second-most abundant CRISPR type [19]. Interestingly, hyperthermophilic archaea of the phylum crenarchaeota are considerably enriched for CRISPR type III effector complexes, which often coexist with CRISPR type I systems, sometimes even sharing crRNAs from the same CRISPR arrays in an organism (see below) [100,101]. Besides RNA targeting and cleavage, type III systems also degrade DNA and, similar to type VI and V systems, can induce collateral cell damage through promiscuous RNA shredding activated by a novel signaling pathway. Thus, type III systems have three different ways to confer immunity to a cell, representing the Swiss army knife of all CRISPR. 

## 3. All Good Things Come in (Type) Threes—A Tripartite Immune Response of CRISPR Type III Systems

Type III systems are further subdivided into subtypes A to F [19,102], of which types III-A through D are the most abundant subtypes with experimentally described variants. Contrarily, the recently defined subtypes III-E and III-F are only found in few microorganisms and remain to be experimentally investigated [19,102]. The most intensely studied type III complexes belong to types III-A, III-D (collectively referred to as Csm) and type III-B (referred to as Cmr). The large subunit Cas10 is generally found in types A-D as well as F and is endowed with cyclase (PALM) and nuclease (HD) domains (sometimes inactivated or missing in certain III-C or III-D variants) (see below, Figure 2). Furthermore, type III-B and type III-A subtypes can be further differentiated by the presence of certain signature genes encoding for the small subunit of the effector complex, namely *cmr5* in III-B, and *csm2* in III-A [68]. Subtype III-D CRISPR-Cas loci share the general effector complex gene composition with those of subtype III-A, however, they encode a distinct *cas5* variant, termed *csx10*, which is a signature gene for this subtype [19,68]. Type III-B complexes often carry the additional subunits Cmr1 and Cmr7, implicated in capping of the target 3′ end and possibly allosteric regulation of RNA cleavage efficiency, respectively [103,104].

Despite those differences, however, type III-A/D and type III-B ribonucleolytic complexes exhibit similar architecture and both bind a class I—specific crRNA (see above and Figure 1 and Figure 2). The crRNA is bound by the backbone consisting of varying numbers of Cas7-subunits (i.e., Cmr4/Csm3), reaching into the Cas5 (Cmr3/Csm2) protein with its 5′ handle. The base of the complex is formed by the Cas10 family protein which interacts with a scaffold consisting of an assembly of small subunits. The latter seem to interact with the target RNA which is hybridized to the crRNA [105,106,107] (Figure 2).

### 3.1. Specific RNA Cleavage

The initial opinion about the target specificity and function of type III systems was split among researchers. Initial in vivo studies of a type III-A system in the mesophilic bacterium *Staphylococcus epidermidis* showed that the system conferred immunity against a plasmid DNA carrying a matching protospacer [108,109], whereas in vitro studies of type III-B systems of the hyperthermophilic archaea *Pyrococcus furiosus* and *Saccharolobus* (previously known as *Sulfolobus*) *solfataricus* [50] and the thermophilic bacterium *Thermus thermophilus,* all demonstrated strong degradation of ssRNA upon crRNA hybridization [101,106,110]. Therefore, despite the striking similarity between Csm and Cmr, they were attributed different target specificities in the early days. Only later, biochemical analysis revealed that type III-A systems universally cleave crRNA-recognized target ssRNA into specific fragment lengths, revealing a 6 nt interval cleavage pattern [106,111,112]. The 6 nt spaced cleavage was later confirmed for the above-mentioned type III-B systems and additionally, for the type III-B of the hyperthermophilic bacterium *Thermotoga maritima* [113,114,115,116,117]. The endoribonucleolytic reaction was found to be executed by Cmr4/Csm3 subunits constituting the middle units of the Cas7 backbone [112,115,117]. A crystal structure of a chimeric Type III complex assembled from subunits of the two archaea *Archaeoglobus fulgidus* and *P. furiosus* bound to a target RNA revealed the reason for this periodic cleavage pattern: every Cmr4 backbone subunit penetrates the crRNA-target RNA duplex with a β-hairpin residue, leading to a distortion at every 6th nucleotide (cf. Figure 2) [107]. Only recently, a very detailed structural investigation of a type III-B system of the archaeon *Sulfolobus islandicus* bound to different target RNAs confirmed these earlier observations, and suggested that additionally to the Cmr4, also the opposing small subunit Cmr5 might play a role in the cleavage reaction [104].

Collectively, these studies clearly confirmed that RNase activity was a universal feature of CRISPR type III complexes and in vivo investigations have proven this to hold true in the living cell, where virus mRNA carrying a protospacer complementary to a host crRNA was efficiently degraded [118,119] (see below).

### 3.2. Unspecific ssDNA Cleavage

The question regarding a potential type III-encoded DNase activity remained, as later in vivo studies observed transcription-dependent DNA cleavage of type III systems [120,121]. Biochemical analysis finally solved the mystery: type III systems cleaved ssDNA promiscuously, once activated by subtle conformational change upon binding of the type III effector complex to a target RNA [101,104,113,114,122,123,124,125]. Cleavage was shown to be mediated by the large subunit Cas10 which is endowed with ssDNA activity conferred by its HD domain [122,126,127] (Figure 2A). Binding of the target mRNA allosterically activates the nuclease domain, as ssDNA cleavage was significantly reduced [127] or totally abolished [123] when the complex was devoid of a target RNA, or when the target RNA disassociated upon cleavage. Thus, the in vitro findings aligned well with the concept of transcription-dependent DNA degradation drawn from in vivo studies: In the cell, the crRNA binds the nascent virus mRNA and similar to a leash, approximates the type III complex to the transcription bubble or any other R- loop nearby [128], which consequently offers a ssDNA substrate for cleavage by Cas10 (Figure 2A). Notably, mechanistic insights of a type III-B complex from *S. islandicus*, that differs from other type III systems by the presence of additional Cmr7 subunits (referred to as Cmr-β), showed that the HD domain also catalyzed unspecific ssRNA cleavage [104], at least in this complex. Upon degradation of the target RNA by the Cas7 backbone, the complex is released and the DNase domain inactivated. Importantly, ssDNA degradation is fully abolished, when the 3′ end of the target RNA was complementary to the repeat- derived 5′ handle of the crRNA (Figure 2B) [104,113,127]. Mutational studies revealed that matches in positions -3, -4, -5 sufficed to inhibit promiscuous ssDNA cleavage by the HD domain in vivo [129] which was confirmed by structural studies showing that the other nucleotides (except for the -2 positions) were inaccessible for Watson-Crick base pairing, as they were either distorted or tightly bound into specific pockets of Cas5 (Figure 2B) [104,107,125]. The 3′ region on the target mRNA is here referred to as protospacer adjacent sequence (PAS) (also termed “PFS” for protospacer flanking sequence or “auto-immunity tag”) and, similar to the above-mentioned PAM, inhibits type III-mediated targeting of the chromosomal CRISPR array as the repeats, being the origin of the 5′ handle, represent a matching PAS. Therefore, even in the rare cases of antisense transcription of a CRISPR array (providing a RNA complementary to the crRNA) [21,110], the HD domain of the type III complex would not be activated owing to the presence of a PAS in the repeat.

### 3.3. Collateral ssRNA Cleavage via cOA Signaling 

Type III genes are often found located proximal to genes coding for proteins endowed with a specific ligand-binding CARF domain (CRISPR — associated Rossmann fold) and an effector domain, most commonly a RNase-specific HEPN domain [130]. While one member of the CARF protein family, Csa3, was shown to regulate spacer acquisition and crRNA synthesis on the transcriptional level in type I-A [48], Csm6 and Csx1 were found to degrade RNA in vitro [131,132,133] and their genetic disruption seemed to decrease the type III immune reaction in vivo [120,134]. However, as neither of these enzymes seemed to physically interact with the type III complex [106,107,112], their interplay and specific role in type III immunity have long remained elusive. Three years ago, two breakthrough studies revealed that CARF-domain enzymes were specifically activated by the type III complex to initiate massive RNA shredding [135,136]. The activation was mediated by the large subunit Cas10 which, additionally to the above-mentioned HD domain, harbors two PALM domains, one carrying a typical GGDD motif, resembling the core domain generally found in nucleotidyl cyclases [137]. Upon crRNA-target RNA binding, the conformational change of the type III effector complex activates the cyclase of this PALM domain which starts to polymerize available ATP into cyclic oligoadenylates (cOA), newly discovered second messenger molecules consisting of three to six AMP monomers (cA3 to cA6). These cyclic molecules in turn allosterically activate the HEPN-ribonucleases Csm6 and Csx1 by binding to their CARF domains [135,136,138,139,140,141], leading to collateral shredding of RNA substrates in the cell. Furthermore, recently the DNA nickase Can1 of *Thermus thermophilus* was also found to be activated by cA4, indicating that cOA-signaling can also impact the DNA level [142]. Similar to type VI systems (see above), cOA-induced RNA shredding was shown to lead to growth arrest which, if continued, could potentially result in cell dormancy or death [143]. Even if the cOA synthesis is deactivated upon degradation of the target mRNA [104,138], already produced cOA can remain in the cell for a longer period, thus keeping HEPN-RNases activated [138]. Only recently, a new protein family, the ring nucleases Crn1 (CRISPR-associated ring nuclease 1), was discovered in the archaeon *Saccharolobus solfataricus* [144] and shown to specifically linearize and therefore inactivate cOA molecules, resetting the cell to a ground state once the virus is defeated by type III systems. While Crn1 ring nucleases seem to be limited to crenarchaeota, a new type, Crn3 (former Csx3) was recently biochemically characterized in the euryarchaeon *A. fulgidus* and found to be widespread in prokaryotes [145]. Furthermore, Csx1/Csm6 ribonucleases, or even Crn-Csx fusion proteins exist which intrinsically degrade the cOA substrates themselves, thus representing self-limiting enzymes independent of a trans-acting ring nuclease [146,147,148,149]. Moreover, in *S. islandicus* REY15A, a membrane-associated DHH-DHHA1 family nuclease (MAD) was recently shown to degrade cOA in vitro and is hypothesized to aid the main cellular ring nuclease in controlling type III immunity by possibly degrading diffused cOA [150]. Intriguingly, specialized forms of ring nucleases were recently found encoded on genomes of archaeal viruses and bacteriophages and were shown to function as anti-CRISPRs to counteract the cOA-induced immune response [151]. Thus, collateral damage by cOA- signaling might constitute the most effective stage of virus defense by type III systems, which is probably especially important when the protospacer is situated in a late-expressed or lowly-expressed viral gene [134,143]. In such scenarios, transcription-dependent ssDNA cleavage of type III is inefficient due to the low transcript number available, leading to an accumulation of the virus in the cell. Yet, cOAs might still be sufficiently produced, causing an interim cell shut down due to promiscuous RNA shredding, thereby preventing completion of the lytic virus cycle and buying time for type III-mediated virus DNA clearance [134,143]. In line with this, a recently discovered anti-CRISPR protein seemed to physically interfere with cOA-mediated virus defense by binding to the type III complex when middle/late viral transcripts were targeted (see below and ref. [152]). 

Importantly, just as the HD domain (see above), the PALM domain and therefore cOA signaling remained inactive when the bound target mRNA contained a PAS hybridizing to the 5′ handle [135,136,138], as it was shown that the induced conformational change leads to blockage of the entrance channel of the cOA substrates [104,125,153].

In summary, one can say that type III complexes pull out all the stops to efficiently curtail a virus spread: First, the virus transcript is recognized and eventually sliced within the protospacer region by the action of the backbone endoribonucleases. Second, when the complex is bound to a target, ssDNA is cleaved in DNA bubbles which is catalyzed by the HD domain of the Cas10 subunit. Third, unspecific RNA shredding by CARF-domain nucleases is activated via secondary molecules synthesized by the PALM domain of Cas10 (Figure 2A). However, all but the first reaction, are allosterically blocked when the 3′ end of the target (PAS) binds the 5′ handle (Figure 2B). Thus, PAS-handle complementarity constitutes an “off switch” for all secondary immune reactions of type III systems, permitting backbone cleavage of the target RNA only.

## 4. CRISPR Research and Application in Archaeal Model Organisms

Shortly after the first discovery of the – back then still enigmatic - regularly spaced CRISPR repeats in the *E. coli* genome in 1987 [154], these arrays were also identified within early sequencing studies of halophilic euryarchaea [155,156]. Whereas initially not experimentally investigated in any bacterium, in vivo studies specifically dedicated to unravelling the physiological role of those repeats were first conducted in *Haloferax volcanii*, where a plasmid engineered with CRISPR repeats was used in transformation assays [156]. Back then, twelve years before the function of CRISPR as an immune system had been resolved, the thereby observed reduction of cell viability and chromosomal content of the polyploid organism was interpreted as a role of the repeats in replicon partitioning [156]. However, from today′s view, Mojica and colleagues might have witnessed CRISPR-mediated self-targeting, triggered by an increased acquisition of chromosomal spacers into the extra CRISPR array [157]—a nowadays well-known phenomenon in halophilic archaea (see below). Thus, even if misinterpreted at the time, one can argue that this study represented the first CRISPR-directed in vivo experiment in a prokaryote. 

The following chapter delineates prominent archaeal model organisms that were used to study and characterize CRISPR mechanisms in archaea. We give an overview about seminal experiments and the current research performed in those organisms highlighting special features of the individual lab strains. A short section is dedicated to the CRISPR applications in archaeal models. Furthermore, a list of the most prominent archaeal lab strains referring to notable publications of CRISPR research performed in the individual organism is presented in Table 1.

### 4.1. Experimental CRISPR Research in Archaeal Model Strains 

#### 4.1.1. Haloarchaea—Attack One’s Own Kind

As genetically tractable and relatively easily cultivatable archaea, halophiles have been used as model organisms in different research fields. However, after the above-mentioned study, there was a long break of using them for CRISPR research, probably because with no identifiable spacer match, halophiles missed out on the boom when CRISPR-targeted viruses were identified [158]. After 17 years of a (non-funded) dry spell, halophilic CRISPR research was revived by characterizing crRNA processing in engineered *Haloferax* CRISPR mutant strains [159,160], as well as CRISPR-mediated DNA interference using plasmid-based invader assays [161,162,163]. Thereby, researchers observed that six different PAMs efficiently triggered an immune response by the type I-B complex (the only effector complex in halophiles), marking the *H. volcanii* Cascade as one of the most versatile type I complexes described [161]. Contrarily, in *Haloarcula hispanica* four PAMs, three of which were distinct from those found in *H. volcanii,* were needed for type I-B plasmid degradation [164]. Besides interference, also spacer acquisition from a halovirus (HHPV-2) has been experimentally shown in infection assays in *H. hispanica*, indicating that it was primed, as additionally to the adaptation cassette, also Cas3 and a partially matching native spacer were required [49]. The thereby established virus-based acquisition assay prompted follow-up studies in this model organism investigating repeat duplication [165], spacer size [166], and crRNA requirements for proper acquisition [167].

By expressing a chromosome-targeting crRNA, it was shown that *H. volcanii* is one of the few prokaryotes that tolerates CRISPR-Cas self-targeting potentially because of a potent microhomology–repair pathway [168], marking it as a distinguished model organism to study CRISPR autoimmunity. For instance, a very recent study demonstrated that overexpression of a self-targeting spacer triggered adaptation of novel spacers collected from the vicinity of the originally targeted chromosomal locus [169]. Apart from the own chromosome, *H. volcanii* and *H. mediterranei* were recently shown to acquire spacers from each other′s chromosomes in mating assays [170], where cells of both species fuse by forming cytoplasmatic bridges [171]. Moreover, when the *H. mediterranei* genome was engineered to be recognized by a native *H. volcanii* spacer in such a crossing experiment, the mating efficiency was decreased. Given the many spacers matching other haloarchaea species found in haloarchaeal genomes [170], this study delivered the experimental proof that CRISPR-mediated cross targeting can shape the gene flux between haloarchaeal species which can be studied in a laboratory set-up.

#### 4.1.2. *Pyrococcus*—Shaping CRISPR Crystals

Advantaged with hyperthermal stability (100 °C) facilitating mechanistic and structural studies of proteins, the hyperthermophilic euryarchaeon *Pyrococcus furiosus* was a pioneer archaeal CRISPR model regarding biochemistry of Cas proteins. Within the early quest to unrevealing processing and architecture of spacer-derived crRNAs (earlier called psiRNA for prokaryotic silencing) [110,172], Cas6 was purified and crystallized from *P. furiosus* [58], leading to its biochemical characterization and the detailed investigation of pre-crRNA binding and cleavage [173,174]. Shortly after that, the first complete prokaryotic type III-B system was isolated from *P. furiosus*, leading to the characterization of the complex composition and the bound crRNAs as well as the first experimental evidence in a prokaryote that type III cleaves RNA in vitro [110]. Additionally, cleavage products of an antisense transcript of a crRNA detected in Northern blots and co-purification of the type III complex verified in vivo activity [175]. These studies prompted the resolution of numerous structures of the different subunits, subcomplexes and entire complexes of the *P. furiosus* type III-B effector ([105,107] and reviewed in [176]) which helped to reveal the molecular details of the ruler-like RNA cleavage mechanism (see above and [107,115,116,177]). Before the type III cOA–signaling pathway was discovered (see above), the accessory protein Csx1 of *P. furiosus* was crystallized [178] and identified to be an adenosine—specific ribonuclease [132]. Recently, it was also demonstrated that like some other CARF-domain nucleases (see above), also Csx1 of *P. furiosus* was endowed with ring nuclease activity, self-inactivating its cOA activators [147]. Apart from additional biochemical studies focusing on the two other type I effector complexes in *P. furiosus* [80,81], in vivo immunity against engineered plasmid invaders was shown for all effector complexes independently by analyzing a plethora of *cas* (and accessory genes) mutants [122,147,179]. Recently *P. furious* has also become a model to study spacer acquisition, revealing that new extrachromosomal spacers are preferentially acquired from broken DNA ends, supplied by plasmids with rolling-circle rather than theta replication [33]. It was further shown that Cas1 and Cas2 alone could acquire spacers in vitro [180], but that Cas4 proteins are essential for acquisition of functional spacers in vivo [45]. Furthermore, by supplying plasmids with partially matching protospacers, primed acquisition could be triggered in *P. furiosus* which was dependent on Cas3 and the type I-B effector complex [181]. 

#### 4.1.3. Methanoarchaea—CRISPR Models on the Fast Lane?

Similar to *P. furiosus*, hyperthermophilic methanogenic lab strains have served to purify and crystallize diverse CRISPR proteins, enabling early biochemical characterization of the type I nuclease Cas3 from *Methanocaldococcus jannaschii* [182] or type III-A nuclease Csm3 of *Methanopyrus kandleri* alone or in a subcomplex with Csm4 [183,184]. Furthermore, the type I backbone subunit Cas8 from *Methanothermobacter thermoautotrophicus* was biochemically characterized and identified to be the PAM recognition factor, essential for interference [163,185]. Array transcription and crRNA processing were studied in the mesophilic models *Methanococcus maripaludis* and *Methanosarcina mazei* from which Cas6 enzymes were purified and characterized in vitro [186,187,188,189]. Furthermore, recent in vivo investigations of *M. mazei* Cas6 mutant strains revealed that only one of the two Cas6 endonucleases executes pre-crRNA maturation of both, type I and type III-adjacent arrays. In vivo studies of CRISPR interference could not be easily performed in native hosts, probably due to the lack of appropriate virus-host systems and assays for methanogens. However, the type I-B system of *M. maripaludis* together with artificial crRNAs was heterologously expressed in *E. coli* demonstrating that, dependent on the flanking PAM, phage lambda infection was reduced to different levels and that efficient interference required Cas8 and Cas3 subunits [190]. 

Most interestingly, methanogens have recently gained attention regarding CRISPR evolution, as Casposons, a sparsely distributed new class of putatively self-synthesizing DNA transposons, were found to be abundant and presumably mobile in *M. mazei* genomes [191]. Recent phylogenetic and biochemical analysis suggest that the transposase (i.e., Casposase) might represent the ancestor of the spacer-integrase Cas1 [192,193], leading to the assumption that CRISPR had evolved from Casposons. Thus, the Casposase of *M. mazei* was recently biochemically and structurally characterized, revealing that it tetramerizes upon target binding and, reminiscent of spacer-acquisition, actively integrated substrates into a preferred target site [194]. Notably, a putative regulator of the Casposase expression was identified in *M. mazei* just now and is published within this special issue [195]. Hence, *Methanosarcina sp.* hold great potential for studying the evolution and mechanisms of CRISPR adaptation. This is not only because they comprise genetically tractable laboratory strains that carry Casposons, but also because they harbor type III-adjacent reverse transcriptases that hypothetically could be involved in spacer acquisition from RNA substrates (see above and ref. [28]). Such RNA-derived spacers could be used to probe for potential archaeal RNA viruses, which haven′t been identified in archaea yet [196]. Furthermore, a recently isolated DNA virus infecting *Methanosarcina sp.* might facilitate CRISPR studies in vivo in native hosts [197]. 

#### 4.1.4. Sulfolobales—The Virus Fighters

The hyperthermophilic archaea of the order Sulfolobales are the most intensely studied archaea regarding the CRISPR system. Many different computational analyses of viral distribution based on spacer tracking have been conducted, array transcription analyzed, and many Cas proteins and CRISPR–related proteins have been biochemically characterized (reviewed in [198,199,200,201,202] and notable references in Table 1). Furthermore, owing to the great number of purified viruses and plasmids [203] and available assays to study virus-host interactions in the culture flask, Sulfolobales lab strains have become distinguished pioneer models for studying CRISPR interference in vivo, some of which will be briefly introduced below. Due to the large number of CRISPR-Cas systems in their genomes, Sulfolobales are also particularly suited to characterize the effects and interactions of multiple CRISPR-Cas systems within one cell.

CRISPR-mediated DNA interference in archaea was first studied in *S. solfataricus* and *S. islandicus* where either a plasmid or a virus was engineered with a cognate protospacer, respectively [204,205]. In plasmid invader assays, where a metabolic gene needed for cell survival was supplied by the protospacer-carrying plasmid, cells only survived upon partial deletion of the native CRISPR locus including the targeting spacer. Thus, by mapping and identifying escape mutations, efficiency of CRISPR-mediated interference could be indirectly determined in this study [204]. In the selection-independent virus approach employing shuttle vectors based on the lysogenic virus SSV1 [206], interference efficiency was directly quantified by counting transfected cells in plaque assays [205]. Both of these strategies were applied in various follow-up in vivo studies to further characterize PAM requirements for type I targeting [207], protospacer-crRNA matches needed for proper interference [129,208], crRNA processing and transcription regulation [207,209] and type III-mediated RNA recognition and interference [119,120]. A significant in vivo finding, already foreshadowing a link between CARF-domain nucleases and the type III-B immune response, was the demonstration of transcription-dependent DNA interference against a plasmid to be dependent on *csx1/csm6* locus and an intact type III-B system in *S. islandicus* [120]. Cells escaping plasmid-targeting arose upon spontaneous deletion of the *csx1/csm6* gene locus, which restored plasmid interference when reintroduced into the mutant cells [120]. Soon after this study, a mutational analysis of protospacer-crRNA hybrids revealed that a match of three distinct base pairs between the PAS–5′ handle sufficiently abolished virus DNA degradation in *S. solfataricus* in vivo [129]. Remarkably, prior to any knowledge of type III collateral damage, this study disclosed the (minimal) sequence requirements for inhibiting secondary type III immune responses in vivo, years before the biochemical determinants were resolved in vitro (see above and ref. [100,104]). Later, researchers found that mutated Cas10-HD-domain type III variants from *S. islandicus* lost their DNA cleavage activity in vitro, but astonishingly, cognate plasmids were still degraded in the respective mutants in vivo, suggesting another type III-mediated interference activity to be in place [124]. 

Type III-mediated degradation of a viral mRNA in vivo in an archaeon was first shown in *S. solfataricus*, where RNA cleavage efficiency could be quantified when the transcribed protospacer (engineered to match a native crRNA) was flanked by a PAS, thereby inhibiting DNA degradation and type III-collateral damage [119]. 40% reduction of protospacer mRNA in the cell was measured using quantitative PCR and Northern blot, and cleavage by the purified *S. solfataricus* type III-B complex was verified in vitro [119]. Analysis of two co-existing type III complexes in *S. islandicus* REY15A using miniCRISPR-based silencing assays in respective mutant strains (see Section 4.2), revealed both complexes to confer differently strong degradation of RNA [118]. Furthermore, a type III complex deficient in the Cmr1 subunit showed decreased RNA and DNA interference in vivo, attributed to decreased target capture efficiency [103,210]. Shortly after the elucidation of the cOA-induced type III signaling pathway in bacteria [135,136], Csx1 of *S. islandicus* was shown to be activated upon binding of its CARF domain to an mRNA adenosine tail [133]. Furthermore, cOA production was characterized for type III-D complexes of *S. solfataricus* [138] and III-B complexes of *S. islandicus* [104,141]. The breakthrough finding of cOA-mopping ring nucleases in *S. solfataricus* probably further fueled the public interest in these model organisms [144]. 

Interestingly, recent in vitro studies of a CRISPR type I-D Cascade from *S. islandicus* LAL14/1 revealed type I-specific dsDNA cleavage as well as ssDNA degradation (see above). Reminiscent of type III-mediated RNA degradation, ssDNA cleavage was found to be ruler-like as governed by the periodically allocated Cas7 subunits of the Cascade backbone. Thus, evolutionary traces of type III systems can be found in the cleavage mechanism of type I-D, suggesting that it constitutes an intermediate between type I and type III systems [27]. This system will be exciting to study in the future, perhaps also because a bacterial type I-D Cascade has recently been successfully applied for targeted mutagenesis in human cells [211].

Apart from studying molecular details of CRISPR interference, infection studies using free virions or environmental virus/plasmid mixes allowed real-time monitoring of the temporal regulation of *cas* genes and CRISPR arrays and the spatiotemporal emergence of viral countermeasures [212,213,214,215,216,217]. Within such experiments, the first two archaeal anti-CRISPR-Cas systems (Acr) encoded by the lytic virus SIRV2 infecting *S. islandicus* were discovered. These were found to either inhibit type I-D or type III-B immunity, respectively, by binding to catalytic complex subunits [152,218]. As mentioned above (see Section 3.3), the functional characterization of Acr-IIIB1 in vivo was particularly important to also understanding the impact of the different mechanisms of type IIIB-immunity on lytic virus infection in *S. islandicus*. Acr-IIIB1 blocked type III cOA signaling only when the protospacer was located on middle/late virus genes, suggesting that collateral damage might be the prevalent immune response acting during the late virus life cycle [152]. Thus, HD-domain-mediated virus DNA cleavage might be predominant when targeting early virus genes, where a higher amount of protospacer transcript is available. 

Very recently, a newly identified ring nuclease was shown to function as Acr in vivo by challenging *S. islandicus* M.16.4, solely carrying the type III system, with a lytic phage (see above). Normally degraded owing to a matching native spacer, the virus could stably infect *S. islandicus* when the Acr was heterologously expressed [151]. Acrs represent exciting subjects to future studies in those model organisms. 

**Table 1 biomolecules-10-01523-t001:** Widely used archaeal model organisms for CRISPR in vitro and in vivo studies. Pioneer studies regarding the respective CRISPR step performed in each model organism are cited.

Archaeal Order	CRISPR Model Organism ^+^	Physiology	CRISPR Types *	CRISPR Steps Studied ^$^	In Vivo Application
Thermococcales	*Pyrococcus furiosus*	hyperthermophilic, anaeorbic	**COM**: Type I-A, Type I-B, Type III-B	Adaptation [33] ^a^, [180] ^b^ Processing [172] ^a^, [58] ^b^, RNA interference [175] ^a^, [110] ^b^ DNA interference [179] ^a^, [122] ^a,b^ cOA signaling ^&^ [147] ^a,b^	
	*Pyrococcus horikoshii*	hyperthermophilic, anaeorbic	**OT3**: Type I-A, Type I-B (x2), Type III-A	Processing [219] ^b^	
	*Thermococcus kodakarensis*	hyperthermophilic, anaeorbic	**KOD1**: Type I-A, Type I-B	Processing [220] ^a^ DNA interference [220] ^a^	CRISPR locus engineered to target invading plasmid [220]
	*Thermococcus onnurineus*	hyperthermophilic, anaeorbic	**NA1**: Type III-A, Type IV-C	DNA interference [126] ^b^ RNA interference [221] ^b^ cOA signaling [146] ^b^	
Methanosarcinales	*Methanosarcina mazei*	mesophilic, anaerobic	**Go1**: Type I-B, Type III-C	Processing [188] ^a,b^ cOA signaling [145] ^c,b^	
	*Methanosarcina acetivorans*	mesophilic, anaerobic	**C2A**: Type I-B, Type III-A		Cas9 genome editing * [222], dCas9 silencing * [223]
Methanococcales	*Methanococcus maripaludis*	mesophilic, anaerobic	**C5**: Type I-B	Processing [186] ^a,b^ DNA interference [190] ^c^	
	*Methanocaldococcus jannaschii*	hyperthermophilic, anaerobic	**DSM 2661**: Type I-A, partial Type III-A	DNA interference [182] ^b^	
Methanobacteriales	*Methanothermobacter thermoautotrophicus*	thermophilic, anaerobic	Type I-B, Type III-A, Type III-C	DNA interference [163] ^b^	
Methanopyrales	*Methanopyrus kandleri*	hyperthermophilic, anaerobic	**AV19**: Type III-A, Type III-B	Processing [224] ^a^	
Halobacteriales	*Haloferax volcanii*	mesophilic, halophilic, aerobic	**DS2**: Type I-B	Adaptation [169] ^a^ Processing [159] ^a^ DNA interference [161] ^a^	CRISPRi: Type I-B gene silencing [225]
	*Haloferax mediterranei*	mesophilic, halophilic, aerobic	**ATCC 33500**: Type I-B	Processing [160] ^a^	
	*Haloarcula hispanica*	mesophilic, halophilic, aerobic	**ATCC 33960**: Type I-B	Adaptation [49] ^a^ DNA interference [164] ^a^	Type I genome editing [226]
Archaeoglobales	*Archaeoglobus fulgidus*	hyperthermophilic, anaerobic	**DSM 4304**: Type I-A (x2), Type III-B	Adaptation [227] ^b^ Processing [228] ^a^ RNA interference [107] ^b^ cOA signaling [145] ^b^	
Sulfolobales ^#^	*Saccharolobus solfataricus*	thermophilic, aerobic	**P1**: Type I-A (x3), Type III-B, Type III-D, partial Type III-B	Adaptation [229] ^a^, [230] ^b^ Processing [231] ^a^, [82] ^b^ DNA Interference [204] ^a^, [232] ^b^ RNA Interference [119] ^a^, [101] ^b^ cOA signaling [151] ^a^, [138] ^b^	Type III gene silencing [119]
	*Saccharolobus islandicus*	thermophilic, aerobic	**REY 15A**: Type I-A, Type III-B (x2),	Adaptation [233] ^a^, Processing [207] ^a^, DNA Interference [204] ^a^, [124] ^b^ RNA Interference [118] ^a^, [124] ^b^, cOAsignaling [141] ^b^	Type III gene silencing [118], Type I genome editing [234], anti-CRISPR based virus editing [235]
	*Sulfolobus acidocaldarius*	thermophilic, aerobic	**DSM 639**: Type I-D, Type III-D	Processing [236] ^a^	Type III gene silencing [237]
Thermoproteales ^#^	*Thermoproteus tenax*	hyperthermophilic, anaerobic	**Kra 1**: Type I-A, Type III-A, partial Type I-A	Processing [238] ^a^ DNA interference [239] ^b^	
	*Pyrobaculum calidifontis*	hyperthermophilic, anaerobic	**JCM 11548**: Type I-A, Type III-B (x2),	Processing [240] ^a^	

^#^ belonging to the crenarchaeota; ^+^ CRISPR types refer to the strain (in bold) with most studies conducted in; * only selected strains are listed, CRISPR types were determined according to refs. [19,68] and CRISPRCasFinder (version CRISPR-Cas++ 1.1.2, [241]); ^$^ referring to pioneer studies covering the respective CRISPR step in strains of the listed species (might contain different strains of the listed species); ^a^ in vivo (Northern blots/RNASeq considered); ^b^ in vitro (cleavage activity of effector complexes or respective signature nucleases); ^c^ in vivo activity shown when heterologously expressed in *E. coli*; ^&^ studies released after cOA-signaling was discovered [135,136] are considered.

In early studies, spacer acquisition in Sulfolobales models could only be induced when applying environmental virus mixtures [229] or specific other co-infecting viruses [233,242], as the presence of particular viruses could specifically trigger spacer acquisition from another component in the mix. For instance, incubation with SMV1 triggered highly selective uptake of spacers exclusively from a conjugative plasmid in *S. solfataricus* or from a co-infecting STSV2 virus in *S. islandicus* [233,242]. In later studies, the transcription factor Csa3a was shown to enhance adaptation *in S. islandicus* [48,243] and spacer acquisition from a single extrachromosomal element could be triggered if it was present in a higher copy number [244,245]. Besides Cas1 and Cas2, Cas4 was shown to regulate spacer acquisition in vivo in *S. islandicus* (see Section 2.1 and ref. [47]) and in vitro in *S. solfataricus* [42,230]. The in vivo dynamics, potentially shaped by Acrs-like mechanisms inactivating spacer acquisition [47], are an interesting field of research, especially with the broad virus selection available for Sulfolobales, and await future studies.

### 4.2. CRISPR Application in Archaeal Models

Shortly after the elucidation of the CRISPR interference mechanisms, many researches focused on exploiting the CRISPR system for genomic engineering. Especially, CRISPR Class II systems have gained tremendous attention and have been engineered as a genome editing tool for setting mutations in bacteria [246] and virtually all eukaryotic model systems including human cell lines (reviewed in [247]). Recently, Cas9 as well as its engineered nuclease-deficient version dCas9 were successfully heterologously expressed in the mesophilic archaeon *Methanosarcina acetivorans* for gene editing and silencing, respectively (Table 1). Upon supplying a repair template that triggered homologous recombination and a crRNA complementary to the target gene, the crRNA-guided Cas9 could induce insertions and deletions in the chromosome of *M. acetivorans*. Repair of the Cas9-induced double strand break without a repair template was only possible when co-expressing a nonhomologous end-joining pathway [222]. The nuclease deficient variant of Cas9 efficiently blocked transcription of desired genes by crRNA-guided binding to the DNA, achieving a silencing efficiency of up to 90% of the *nif* genes involved in nitrogen fixation [223].

Contrarily to *M. acetivorans*, stable expression of the commonly used Cas9 of the mesophilic bacterium *Streoptococcus pyogenes* [83] in hyperthermophilic or halophilic archaea could not be achieved due to instability of the enzyme [248]. However, in those archaeal models, endogenous type I and type III systems can be efficiently hijacked for gene silencing and genome editing (Table 1). *H. volcanii* mutants, that carry an endogenous DNA-targeting Type I-B complex deficient in Cas3 nuclease activity can be used for gene silencing via transcription blocking (i.e., CRISPR interference), similarly to dCas9 [225]. CRISPRi efficiency can be increased by preventing the occupation of the available dCascade complexes by endogenous crRNAs which can be achieved by deleting either the native CRISPR arrays or the processing gene *cas6*, respectively. To ensure proper processing of the artificial crRNA in the latter approach, the crRNA must be flanked by t-elements that are recognized and processed by endogenous tRNases, generating a mature crRNA [225]. CRISPRi was successfully used in *H. volcanii* to silence non-essential and essential genes, achieving 78% knockdown of the essential RNase P [225]. Besides gene silencing, in *S. islandicus* as well as *H. hispanica*, the native intact type I system was successfully exploited for genome editing by supplying a helper plasmid and an artificial crRNA targeting the chromosomal locus to be edited [226,234]. As for Cas9, silencing or editing via a type I system requires a PAM in the target sequence.

Interestingly, an innovative virus editing technology makes use of an anti-CRISPR system, conferring immunity to I-D-mediated DNA interference in *S. islandicus* LAL 14/1, as selection marker for targeted gene knockouts in virus derivates (deficient of an Acr) in vivo [235].

A native CRISPR type III system can be repurposed for posttranscriptional silencing of host genes in *S. solfataricus* [119,249], *S. islandicus* [118,250] and *S. acidocaldarius* [237]. For type III-mediated silencing, crRNAs are heterologously expressed from a miniCR vector, incorporated into a native CRISPR type III endonuclease and guided to complementary loci on the mRNA of a desired gene which is subsequently cleaved [118,119]. The protospacer on the mRNA requires a PAS in order keep the collateral damage immune response and unspecific ssDNA cleavage of type III systems inactivated (see above and [119]). By increasing the number of expressed crRNAs, we could gradually increase the knockdown levels, which were stably maintained over the course of growth in *S. solfataricus* [249,251]. In theory, this technology can readily be applied in any genetically accessible organism carrying a type III system, as no genetic manipulation of the complex is required beforehand. Besides some non-essential genes that could be silenced to almost 100% [118,249,250], we have recently applied the type III-mediated knockdown on essential genes of different functional categories, including cell division, transcription, cell wall biogenesis and translation [237,251,252]. We found that dependent on the gene, a maximum silencing efficiency of 40–75% was achieved and could not be exceeded [237]. Within this range, the silencing effect was stable and specific phenotypes could be analyzed in vivo, allowing functional characterization of the respective gene [251]. Higher silencing levels conferred by stronger spacers (or an increased numbers of otherwise stable spacers) were not tolerated, leading to precise excision of those spacers from the miniCRISPR array. Thus, this suggests the presence of a probably CRISPR-linked mechanism to eradicate deleterious spacers [198,237].

## 5. A Hot Fuzz: CRISPR Immunity in Sulfolobales

Members of the order Sulfolobales are thermoacidophilic crenarchaea, belonging to the TACK superphylum within the archaea. Almost all thrive at around 80 °C and a pH of 3 on organic carbon sources under aerobic conditions and are found in high temperature, mostly terrestrial environments, such as solfataric and volcanic hot springs in Iceland, Kamchatka or Yellowstone National Park. Owing to the extremophilic life style, Sulfolobales have become literally “hot” objects of study regarding industrial applications, biochemistry and also evolution as they potentially could have withstood the harsh environments on an early Earth [253]. Moreover, Sulfolobales are known to share their environments with many viruses [254]. The Sulfolobales viruses that have been described so far belong to six different archaeal virus families and are characterized by an impressive diversity regarding their morphologies, as well as genome structures and life cycles [203,254]. While viruses of bacteria are mostly lytic, the majority of archaeal viruses rather seem to persist in their host cells in a stable carrier state [255], which sometimes is beneficial for the host [256]. Similar to prophage lambda, some temperate archaeal viruses can be activated by various stimuli, such as the well described *Fuselloviridae* infecting Sulfolobales, however they do not always cause cell lysis after induction of virion production. A recent study on host-virus interactions of geographically separated, natural *S. islandicus* populations with SSVs and SIRVs suggests that the CRISPR-Cas immunity is more diversified in response to lytic viruses and free virions compared to non-lytic and integrated viruses [257]. In light of the enormous diversity of viruses, but also of large numbers of IS elements in Sulfolobales, it is not astonishing that they are equipped with extensive CRISPR-Cas systems, which makes them particularly interesting study objects for this field. We here present a thorough comparative genomic analysis on the distribution and abundance of CRISPR-Cas systems in all sequenced genomes available to date.

### 5.1. Distribution of CRISPR Types in Sulfolobales

Altogether, the 38 fully sequenced representative Sulfolobales genomes currently available harbor 124 individual CRISPR-Cas loci assigned to two different CRISPR-Cas types, namely type I (52 loci, 42%) and type III (72 loci, 58%) (Figure 3, Upper panel). The derivative strains of *S. solfataricus* SULA and *Metallosphaera sedula* DSM 5348 (15 strains in total) generated via adaptive laboratory evolution were excluded from the dataset to not skew the analysis, as they exhibit similar CRISPR-Cas systems and spacers as the parental strains. Amongst all type I CRISPR-Cas loci present in Sulfolobales, I-A is by far the most prevalent subtype (33x), followed by I-D (14x) and I-B (5x) (Figure 3, Lower panel). While type I-B systems have been shown to target DNA in several other archaea [161,179], type I-A and I-D mediated DNA targeting was experimentally verified in members of the Sulfolobales [27,204,207,218].

The majority of type III CRISPR-Cas loci in Sulfolobales consists of subtypes III-D (32x) and III-B (22x), whereas in comparison the third designated subtype III-A represents a rather small fraction (3x). Interestingly, a considerable fraction (15x) of the detected CRISPR-Cas loci constitute a distinct variant of type III systems exclusively found in Sulfolobales, predominantly in different strains of *S. acidocaldarius* and *S. islandicus* (see Figure 3 lower panel, Figure 4, and see [19]). These complexes comprise divergent Cas10 and Cas5 subunits as well as a putative additional subunit of unknown function termed Csx26 and need yet to be experimentally investigated [19].

All members of the Sulfolobales possess at least one complete type I or type III CRISPR-Cas system, the sole exception being *Stygiolobus azoricus* FC6, which only contains an incomplete subtype III-D locus (Figure 4, Appendix A). While *S. solfataricus* SULA and its derivative strains only harbor a single subtype I-A system, *S. acidocaldarius* Y14_18-5 and *Acidianus ambivalens* LEI10, in contrast, both exclusively harbor one CRISPR-Cas system of subtype III-D. Apart from the exceptions mentioned above, all other Sulfolobales genomes encode more than one CRISPR-Cas system, and in those genomes type I and type III systems always co-exist (Figure 4, Appendix A). *S. solfataricus* P2 is the current record holder with a total of six complete CRISPR-Cas loci, more precisely, three loci of subtype I-A, two of subtype III-B and one locus of subtype III-D (Figure 4, Appendix A).

Notably, *S. azoricus* FC6 and *A. ambivalens* LEI10 represent the only two genomes which do not contain any or a seemingly non-functional adaptation cassette (internal stop in *cas1*, see Appendix A), respectively, and hence in those organisms, spacer acquisition is probably impaired (Figure 4, Appendix A). Furthermore, we could not identify a *cas6* in the *S. azoricus* FC6 genome, indicating that this organism might not be able to process pre-crRNAs. Thus, despite harboring three CRISPR arrays (Figure 5A), *S. azoricus* FC6 might be the only representative of the Sulfolobales without functional CRISPR–Cas immunity, as it lacks all crucial genes for spacer acquisition, maturation as well as a complete interference module (Figure 4). All other members of the Sulfolobales, additionally to complete effector complexes (see above), possess at least one functional gene set for spacer acquisition as well as at least one *cas6* and therefore should be capable of performing all steps of CRISPR immunity (Figure 4, Appendix A, cf. Figure 1).

The accessory genes *csx1*, which encode indiscriminatory ssRNA nucleases playing a crucial role in the CRISPR type III immune response (see above), could be readily identified in almost all Sulfolobales genomes. As described above, cOAs produced by type III effector complexes serve as activators for Csx1/Csm6, and in turn those signaling molecules are degraded by ring nucleases Crn allowing the CRISPR immune response to return to ground state (see above). Thus, the type III effector complex, Csx1/Csm6 and Crn should co-occur in order to govern the controlled catalytic cycle of the type III immune response [259]. Indeed, we generally find all three components in one genome, however, there are exceptions to this rule (Figure 4). *S. solfataricus* SULA (and derivative strains) contain both *csx1/csm6* ribonucleases and ring nucleases, but lack a type III system indicating that these might have functions beyond type III immunity [260]. Contrarily, *S. islandicus* Y.G.57.14 which encodes four type III systems (3x subtype III-B, 1x type III unclassified), thereby representing the highest detected number of type III systems within our dataset, marks the only genome without an identifiable *csx1*/*csm6* gene, suggesting no cOA-driven collateral damage to be in place. Furthermore, for *S. islandicus* strains (M.16.4, and Y.G.57.14) we could not identify any ring nuclease, although both are equipped with type III effectors. Although *S. islandicus* M.16.4 encodes for the promiscuous RNA shredding Csx1, its antagonistic Crn seems to be dysfunctional as it is disrupted by an internal stop codon (Figure 4 and Appendix A). Thus, if type III effectors as well as CARF-domain nucleases are active in those strains, it remains unclear how cOAs levels are controlled. One explanation could be, that some CARF-domain nucleases are self-limiting and degrade cOAs by themselves, as has been observed for other organisms (see above and refs. [146,148,149]). It could also be that sometimes collateral damage is regulated in trans, through other genetic elements like Crn2 ring nucleases encoded by archaeal viruses, such as STIV which is abundant in Sulfolobales (see above and ref. [151]). For two other *S. islandicus* strains (M.14.25 and M.16.27) only two ring nuclease gene copies but no *csx1* gene could be identified (Figure 4). It should be noted that some ring nucleases in our analysis were previously assigned to the Csx1/Csm6 family [19], however they do not seem to contain a bona fide HEPN domain. Moreover, biochemically characterized representatives encoded by *S. solfataricus* P2 did not show cA4-stimulated RNase activity however did exhibit cA4 degradation activity in vitro [144].

CRISPR regulators (*casR*) shown to induce (Csa3a) or repress (Csa3b) expression of adaptation cassettes/CRISPR arrays and effector complexes, respectively [48,261], were found in all genomes, except *S. solfataricus* SULA and derivates, *Sulfuracidifex tepidarius* as well as *S. azoricus* FC6 (Figure 4).

### 5.2. CRISPR Arrays and Virus Matches in Sulfolobales

In general, Sulfolobales harbor several CRISPR arrays, ranging from only two in *M. cuprina* Ar-4, *Sulfodiicoccus acidophilus* HS-1, *S. acidocladarius* Y14 18-5 and some *S. islandicus* strains, to up to ten in *A. sulfidivorans* JP7 and *S. tepidarius* IC-006 and IC-007 (Figure 5A, Appendix A). Although containing only two CRISPR arrays, *S. acidophilus* HS-1 exhibits the third highest total number of spacers (447), as both arrays encompass more than 200 spacers each, thereby also representing the by far longest CRISPR arrays amongst the Sulfolobales. This large total number of spacers is only exceeded by unclassified *Saccharolobus sp.* A20 and *Sulfurisphaera tokodaii* 7, harboring 463 and 454 spacers, respectively, partitioned between 6 CRISPR arrays each. In comparison, *S. acidocaldarius* Y14 18-5 only harbors 14 spacers within its two CRISPR arrays altogether, and hence clearly contains the lowest total number of spacers (Figure 5A, Appendix A).

**Figure 4 biomolecules-10-01523-f004:**
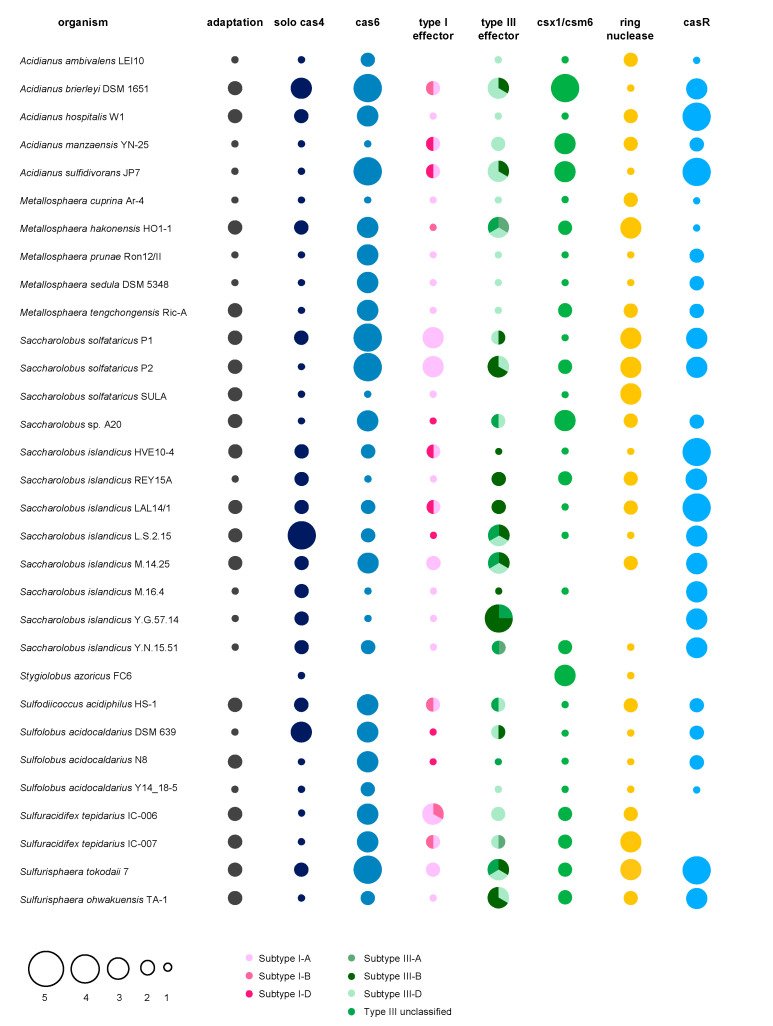
Distribution of CRISPR-Cas (sub) types and accessory genes in Sulfolobales genomes. The bubble plot shows the distribution and abundance of genes encoding proteins and protein complexes partaking in the CRISPR-Cas immune response in Sulfolobales genomes. The abundances of adaptation cassettes (*cas1-cas2*, frequently also *cas4*), *cas6* (crRNA processing) and CRISPR-Cas effectors (interference) are depicted by size; abundances of the different subtypes of CRISPR-Cas effectors are indicated by different color shadings. Some adaptation cassettes contain either a *cas1* or *cas2* gene with internal stop codon or frameshift, highlighted in Appendix A. Additionally, abundances of solo *cas4* (not encoded within 15 ORFs with respect to other adaptation genes) and accessory genes *csx1*/*csm6*, *crn* (ring nuclease), *casR* (specific transcriptional regulator) are illustrated by bubbles in corresponding sizes. In cases with great overlap between different strains of the same species, only one genome is shown as a representative for all similar strains (e.g., *S. solfataricus* SULA and derivative strains). The analysis is based on data published in [19], and/or obtained by using programs CRISPRminer (version 1, [258]), CRISPRCasFinder (version CRISPR-Cas++ 1.1.2, [241]), and BLAST ([262,263]); BLAST analysis of Crn and Csx1 was performed based on amino acid sequences of the respective biochemically and structurally characterized proteins [139,144].

Collectively, the 38 Sulfolobales genomes accommodate over 10,400 spacers, and for approximately 6.7% of those spacers (partially) matching protospacers could be identified in the genomes of known Sulfolobales viruses (see Appendix A). This value is comparable to previous analyses which were based on the entirety of spacers from all publicly available prokaryotic genomes or the CRISPRome of a natural population of Sulfolobales (7 and 6% in ref [24,264], respectively). In general, hyperthermophilic archaea are especially enriched in CRISPR-Cas systems, however readily identifiable spacer matches to viral genomes are comparably scarce [19,24]. As mentioned above, this observation possibly reflects the enormous variety of viruses which yet remain to be discovered [24]. While *S. solfataricus* P1 and P2 exhibit the highest absolute numbers of virus matching spacers (namely 83 and 91 spacers, respectively), two *S. islandicus* strains, Y.G.57.14 and Y.N.15.51, exhibit the highest relative proportion of spacers (around 24 and 30%, respectively) with identifiable matching protospacers in viral genomes (Figure 5A, Appendix A). Generally, the top three targeted virus species (highest numbers of protospacers found in their genomes) are *Sulfolobus islandicus rod-shaped virus* (SIRV), *Sulfolobus monocaudavirus* (SMV) and *Acidianus two-tailed virus* (ATV) (Figure 5B).

### 5.3. A General Scenario for CRISPR Interference in a Sulfolobales Cell

According to our analysis and the current literature, we can draw simplified scenarios with three possible levels of how CRISPR interference against a cognate virus (i.e., carrying a protospacer that perfectly matches a native CRISPR spacer) could be achieved in a Sulfolobales cell. These scenarios do not consider variables, such as viral toxins, that might have an additional impact on virus-host interactions in nature. We reason that the efficiency of CRISPR interference largely depends on protospacer flanking motifs, such as PAM or PAS, and the transcription of the protospacer.

**Figure 5 biomolecules-10-01523-f005:**
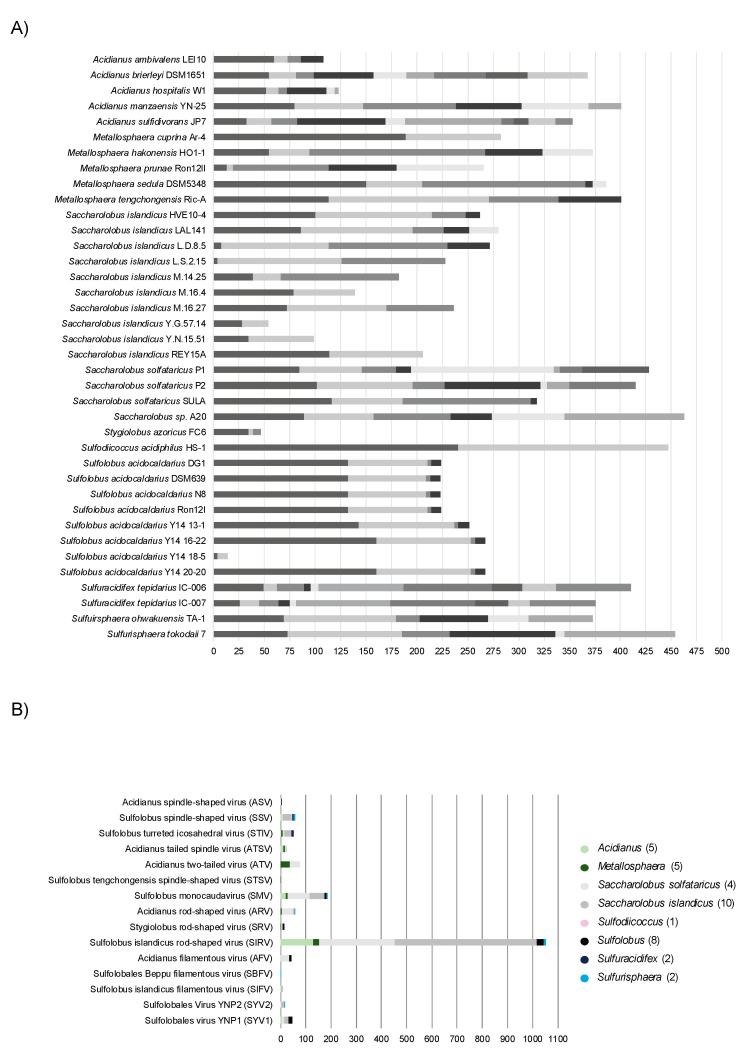
CRISPR array, spacers and virus matches in Sulfolobales genera. (**A**) The number of arrays (depicted in different color shadings) and the number of spacers per array are shown for representative genomes of the Sulfolobales. (**B**) The stacked bar plot shows the total amount of spacers identified on the genus level that (partially) match protospacers carried on genomes of viruses associated with the Sulfolobales. The data (CRISPR arrays and spacer) for the analyses were retrieved using programs CRISPRCasFinder (CRISPR-Cas++ 1.1.2, [241]) and orientation of CRISPR arrays was determined using CRISPRstrand (implemented in CRISPRmap v1.3.0-2013, [265,266]). Virus matches were identified using BLAST+ (version 2.10.0, [267]), spacers were blasted against the NCBI viral genomic RefSeq database [268] specifying the following parameters: word size = 8, e-value ≤ 0.01. Results were filtered for a query coverage ≥ 85% (calculated by dividing the alignment length by the query length) and allowing for a maximum number of 5 mismatches (see Appendix A), duplicate hits of a spacer to the same virus were removed.

In the presence of a functional PAM sequence, the type I Cascade can recognize and degrade the DNA of the infecting virus (Figure 6, upper panel “efficient”). Previous interference studies in *S. islandicus* mutants only carrying a type I-A system as sole effector module have shown that such type I-A—mediated interference can be sufficient to efficiently eradicate an invading plasmid [207]. However, in a theoretical scenario where the type I system alone could not provide efficient DNA interference, possibly due to invasion of an overbearing amount of virus copies, masking of the protospacer/PAM or due to type I-A anti-CRISPRs [215], the type III complex could function as a “backup” system to confer immunity (Figure 6, upper panel, “inefficient”) [269]. crRNAs transcribed from different CRISPR arrays were shown to incorporate into both, type I and type III systems in *S. solfataricus*, substantiating that type III can utilize the same crRNAs as type I [82,101]. This however could only occur if the protospacer is transcribed and its mRNA (or antisense RNA) matches the cognate crRNA (Figure 6, upper panel “inefficient”). As a functional PAM always mismatches the 5′ handle of the crRNA, all three immune functions of the type III system (RNA degradation, HD-mediated unspecific ssDNA cleavage and cOA-signaling-dependent collateral RNA damage) would then be activated, efficiently eradicating the infecting virus. The fate of the cell would depend on the efficiency of the ring nuclease resetting the cell to a ground state after the virus has been defeated (see above and refs [138,144]).

In a second scenario, a protospacer is flanked by a PAS, matching the 5′ handle of the native crRNA. As the protospacer would not be recognized by the type I complex on DNA level due to the missing PAM, the virus can efficiently infect the cell and propagate, leading to virus spreading in the population (Figure 6, lower right panel “untranscribed PS”). However, if the protospacer is transcribed and the transcript can hybridize to the native crRNA, the type III complex can be activated to degrade the virus RNA (Figure 6, lower right panel “PS transcript”). As the handle – PAS match allosterically inhibits activation of secondary immune responses, the type III system would only silence the virus by specifically degrading its RNA. This could lead to a stabilization of the intracellular virus number (i.e., prophage stage) while inhibiting the propagation of the virus [121]. Indeed, expression of a virus mRNA carrying a cognate protospacer flanked by a PAS in *S. solfataricus* led to stable virus DNA copy numbers, whereas the targeted mRNA levels were reduced [249]. Thus, in scenario 2, the chance is highest for the virus to persist in the population.

If an infecting virus carries a protospacer without a flanking region that is similar to a PAM/PAS, the type I system would not be triggered (Figure 6, lower left panel “untranscribed PS”). If the protospacer is transcribed, survival of the cell would depend solely on the proper activity of the type III complex for eradication of the virus, and the efficiency of the ring nuclease in degrading residual cOAs (Figure 6, lower left panel “PS transcript”).

We conclude that in Sulfolobales harboring type I and type III systems, viruses carrying a protospacer with a PAM would most likely be efficiently degraded and eradicated from the population, whereas protospacers flanked with a PAS would have the highest probability to persist in a carrier or prophage state. However, as mentioned above, these dynamics can change under different environmental conditions and probably largely depend on the nature of the infecting element—factors we did not consider here. For instance, a silenced virus could theoretically still kill the host upon expression of a harmful protein.

Although it is quite probable that not all functions and benefits of CRISPR immunity have been discovered yet, it becomes clear that the diversity of viruses is reflected in an impressive diversity of CRISPR-Cas systems in the Sulfolobales. The sophisticated immune system in turn does not necessarily always result in eradication of viruses but may also foster co-existences that are known to be beneficial to natural microbial populations, as they increase genetic diversity and exchange.

## Figures and Tables

**Figure 1 biomolecules-10-01523-f001:**
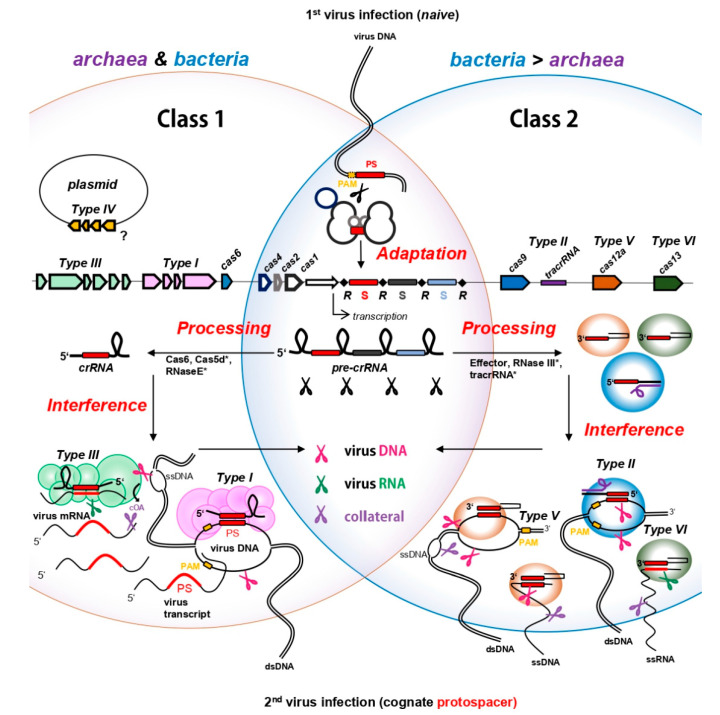
CRISPR types in prokaryotes and their mechanisms of action. Class 1 systems (including types I, III, and IV) are prevalent in both archaea and bacteria, whereas Class 2 systems (including types II, V and VI) are found almost exclusively in bacteria. Mechanisms and genes in the overlapping region are shared by the two classes. The different CRISPR steps are indicated in red and explained in the main text. Both classes contain effector complexes that can perform all three types of interference: virus DNA interference (indicated by pink scissors), virus RNA interference (indicated by green scissors) and collateral damage (i.e., promiscuous cleavage of host and virus DNA/RNA, indicated by purple scissors). “S” refers to “spacer”, “R” refers to “repeat”. * Asterisks refer to enzymes that are not involved in processing of all classes: Cas5d replaces Cas6 function in type I-C [25]; RNaseE is involved in processing in a type III-B system [26]; RNase III processes pre-crRNA in type II; tracrRNA is needed in type II and certain subtypes of type V (see text). Type I-D targets ssDNA and dsDNA (not shown) [27]; In type II systems, Csn2 is also involved in spacer acquisition (not shown); spacer acquisition from RNA via RTs is not shown [28]; ssRNA targeting of the type V-G subtype is not shown [29] (see text).

**Figure 2 biomolecules-10-01523-f002:**
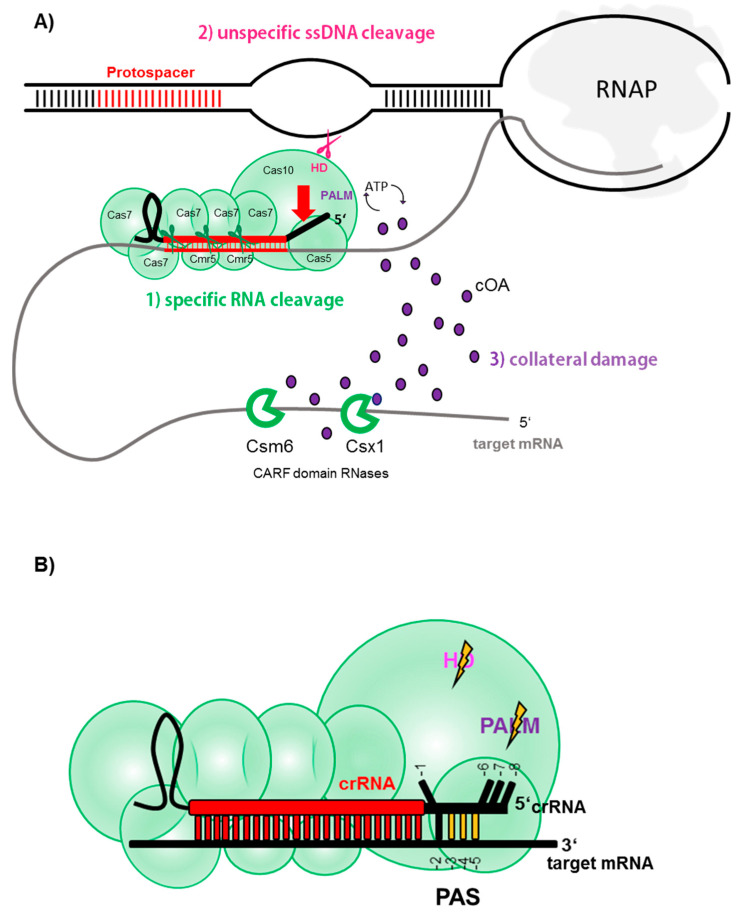
Activated and deactivated immune response of CRISPR type III complex. (**A**) The three phases of type III–mediated immunity are activated when a nascent mRNA is recognized by a crRNA in a type III complex. (1) Specific RNA cleavage: The Cas7 backbone cleaves the mRNA specifically at the protospacer region. (2) Unspecific ssDNA cleavage: When crRNA 5′ handle and PAS are unpaired (red arrow), the HD domain (pink) of the large subunit Cas10 is activated and mediates sequence-unspecific ssDNA cleavage of nearby DNA bubbles. (3) Collateral RNA shredding: As in the case for ssDNA activation, the PALM domain (violet) in the large subunit Cas10 is activated if the 5′ handle is unpaired to the PAS (red arrow). The PALM domain converts ATP into cyclic oligoadenylates which bind to the CARF domain of RNases, such as Csm6 and Csx1, thereby activating nonspecific RNA shredding. (**B**) Base pairing between the 5′ handle of the crRNA and the 3′ PAS. Complementarity in regions -3, -4, -5 deactivates HD and PALM domain activity, thereby only allowing specific RNA cleavage mediated by the backbone of the complex. Nucleotides in position -1, -6, -7, -8 do not contribute to base pairing, as they are distorted (-1), or tightly bound into specific pockets (-6, -7, -8).

**Figure 3 biomolecules-10-01523-f003:**
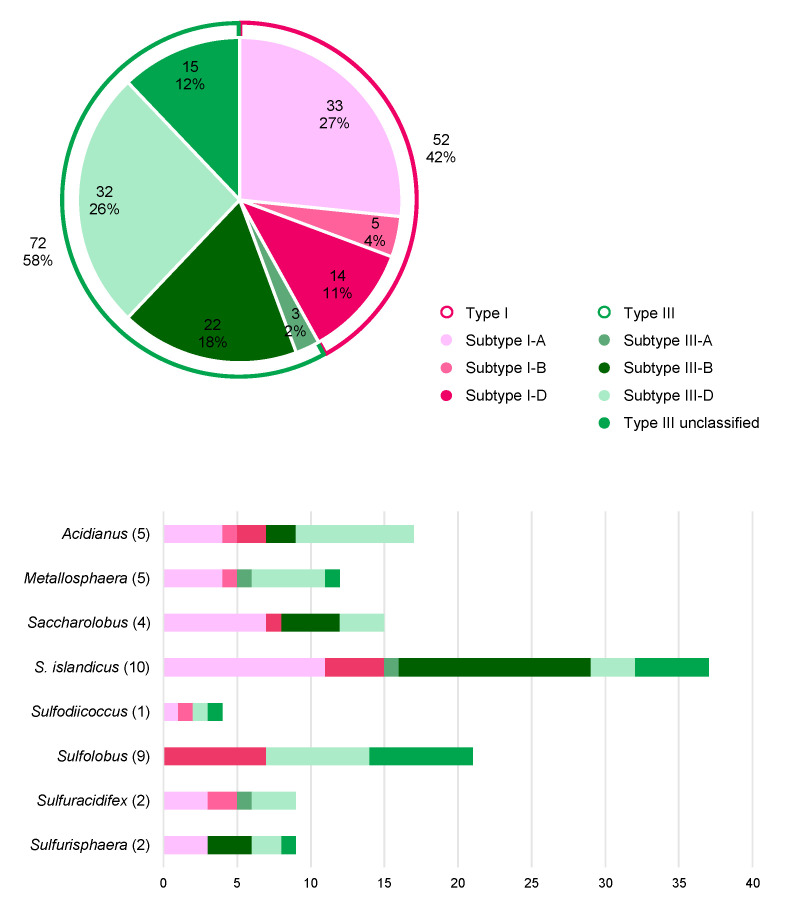
Presence of specific CRISPR-Cas (sub)types and their abundances among Sulfolobales genera. Overall abundances of CRISPR-Cas types and the specific subtypes within the general types I and III found in the order Sulfolobales (Upper panel). The bar chart shows the distribution of the CRISPR-Cas subtypes throughout the genera within Sulfolobales (Lower panel). The number of genomes included in the analysis for each genus is given in brackets. The analysis is based on data published in ref. [19] and/or obtained by using programs CRISPRminer (version 1, [258]), and CRISPRCasFinder (version CRISPR-Cas++ 1.1.2, [241]).

**Figure 6 biomolecules-10-01523-f006:**
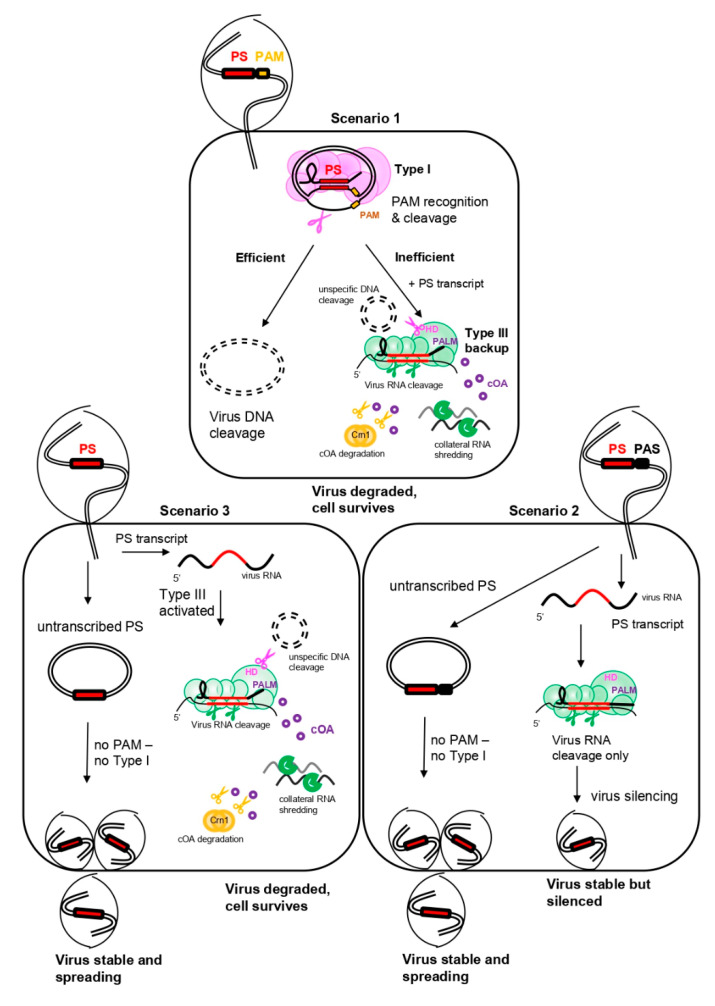
Three scenarios of anti-virus immunity in a Sulfolobales cell. Scenario 1 (PAM) would be the most efficient to degrade a virus in a Sulfolobales cell, whereas scenario 3 (no PAM, no PAS) could lead to virus spreading and/or cell death. Scenario 2 (PAS) is the only scenario that could lead to virus silencing on RNA level and, consequently, persistence of the virus in the population.

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
