# Peer review of "Heavily Armed Ancestors: CRISPR Immunity and Applications in Archaea with a Comparative Analysis of CRISPR Types in Sulfolobales"

_biomolecules, 2020, doi:10.3390/biom10111523_

Round 1

Reviewer 1 Report

This review by Zink et al., is an exhaustive compilation of all the publications available on CRISPR-Cas systems in Archaea. Although there have been many reviews on CRISPR-Cas, here special focus is given to the, sometimes ignored, Archaea domain. As part of the article, the authors have also performed new analysis of the CRISPR-Cas systems in Archaea, which will definitely be useful to the community. Specifically, the distribution and possible co-existence/inter-dependence of type I and type III systems along with the distribution of Csx1 and ring nucleases among the members of Sulfolobales. In addition to this, they have also identified spacer matches to viral genomes. I must take this opportunity to thank the authors for their effort into making such an essential Archaea specific review. Having said that, we have to remember that many will refer to this review in the future, so for their benefit I do have some suggestions that I would like the authors to incorporate to make the review more impactful.

  1. In the sentence “These “innate” defense systems represent a first barrier against exogenous infection but are often inefficient or overcome by the co-evolving virus” do the authors imply that toxin antitoxin system, BREX, DISARM etc are the innate first barrier but not CRISPR-Cas? Please clarify.
  2. Page 4, “organism-specific recognition motif termed PAM” it is my understanding that the PAMs are subtype (probably even repeat sequence) specific and do not vary since the residues in the PAM recognition domain remains conserved across organisms.
  3. Page 5, could you please add a few sentences on the recent identification of internal translation sites for the small subunits within the large subunit transcripts identified as part of the recent type I-D publications. (https://doi.org/10.1101/2020.03.14.991976, https://doi.org/10.1101/2020.04.18.045682 and PMID: 32960267). This is an important development, relavent to most subtypes of type I CRISPR-Cas.
  4. Page 9, the title “Collateral ssDNA damage” is misleading. Cleavage of viral DNA cannot be referred to as collateral damage. The ssDNA activity of type III complexes are spatiotemporally triggered in order to specifically target viral DNA, if they collaterally damage host DNA then type III activity would lead to cell suicide which is not the case. This is unlike the Csx1/Csm6 RNase, which can be referred to as collateral ssRNA cleavage, because its activity has been shown to cleave host transcripts (PMID: 30692669).
  5. Page 9, “Notably, mechanistical insights of the type III-B complex form S. islandicus showed that the HD domain also catalyzed unspecific ssRNA cleavage”. This is a feature unique to cmr-beta complex and should not be generalized to subtype III-B.
  6. Page 9/10, it is a pity that the role of viral ssDNA cleavage and collateral viral/host transcript cleavage in the in vivo early and late viral gene targeting (high and low plasmid transcripts) is not described (type III-A PMID: 26853474 and 30692669, type III-B PMID: 31564454). It is a very interesting and novel concept.
  7. Page 13, sentence “Later, contradictive findings showing that isolated Cas10-HD-domain type III variants from S. islandicus lost…..”. Please note that only the HD domain mutant lost its ability to cleave DNA in vitro but in vivo the mutant exhibited reduced transformation efficiency presumably due to the activity of Csx1, which was not known at the time.
  8. Page 13, missing study on in vivo characterization of type III-B system specifically the role of HEPN and HD domains during lytic virus infection (mentioned above, comment 6).
  9. Table 1, citation 239 (anti-CRISPR based genome editing) is not for islandicus Rey15A, it is specific to S. islandicus LAL14/1 which encodes one type I-A, one type I-D and two type III-B (cmr-alpha and cmr-gamma) CRISPR-Cas systems.
  10. Page 17, If possible, can you please elaborate on the significance type I and type III systems co-occurring in Sulfolobales?
  11. Figure 4, The CRISPR-Cas composition of islandicus LAL14/1 and S. islandicus HVE10/4 are identical, yet your figure shows differences in type III.
  12. Page 22/23, the second scenario, virus stabilization, from the view of the host it is not possible. Does this mean that lysis cannot be induced when the host carries a SSV1-protospacer with a matching PAS? Stabilization of SSV1 in the host (citation 250) can only be co-related to complementary PAS, there might be additional unknown factors involved. Targeting a single gene with a matching PAS does not mean other, potentially harmful, viral genes are not expressed. In this case, the virus might not complete its life cycle but the cell will not survive. Please consider revising your hypothesis.

Supplementary File

  1. In the arrays tab, the repeat sequences in individual arrays of individual Archaeal species are listed. This is highly informative, so I hope the authors will correct the sequences and adjust for the array orientation. Repeat sequences among different arrays are some times similar but when the orientation is not corrected, they can look different. For example in islandicus Rey15A reverse complement of CTTTCAATTCTATAGTAGATTAGC is GCTAATCTACTATAGAATTGAAAG, which is identical to the second array listed as part of Rey15A. I hope the authors will make the changes for all species listed as this is an important detail.
  2. In the virus matches tab, virus matches seem to be incomplete. I very much appreciate the effort to meticulously look for spacer matches among viruses, but the list seems to be incomplete. For examples SIFV and SIFV2 match to islandicus LAL14/1 CRISPR array 1 spacer 80 is missing (Sequence : TAGAAGATGTGCAACAGTTATGTACACAGTATTGTACAAA, 95% match). If you could have a second look and add any missing matches it would enhance the significance of the data presented.
  3. In the csx1-csm6-crn tab, a description for * is missing.

Other comments

  1. In section 2.3, a few lines on the assembly of Cas proteins on the mature crRNA and the necessity of the 5’ handle would be ideal.
  2. genome rendering: comma missing between genome and rendering.
  3. Page 5, please cite PMID: 30284045 and 27582008 among citations [70-76].
  4. Page 6, Protospacer flanking site is abbreviated incorrectly. It is PFS, not PSF.
  5. Use of the word similarly in multiple locations might be in correct please revise.
  6. Page 9, check typo. Mechanistic insights of type III-B systems from S. islandicus.
  7. Page 10, please cite newly identified membrane associated ring nuclease (PMID: 32937129)
  8. Page16, there is also in vitro evidence for type I-D in Sulfolobus.
  9. Page 20, the sentence “While solfataricus P1 and P2 exhibit the highest absolute…… (Fig. 5A)”. Figure 5A does not clearly indicate identifiable protospacer matches, I think reference to supplemental table is missing.
  10. Page 22, in the sentence “The faith of the cell would depend…..”, you mean the fate of the cell….
  11. Double citations 251,252 and 253,254.

Reviewer 2 Report

This is a timely and comprehensive review of CRISPR-Cas systems in archaea. The illustrations are excellent and the text well written. I have only minor comments (see below).

Minor comments

First, general comment: CRISPR systems need cas genes and therefore usually they are referred to as CRISPR-Cas systems. Here the terms CRISPR-Cas systems and CRISPR systems are used interchangeably.  The authors should either have a uniform style or they should state that they use both terms in one of the opening paragraphs. 

Abstract

“In the last chapter” – should be “In the last section”

Section 1

41 “the particles of prokaryotes (i.e. Archaea and Bacteria) infecting viruses” should be “the particles of

Prokaryote-infecting (i.e. those infecting Archaea or Bacteria))”

78 “resistance against the virus [16]” should be resistance against the virus [16]"

101 “Class 1 systems (including types I, III, and IV) are prevalent in archaea” should be “Class 1 systems (including types I, III, and IV) are prevalent in archaea and bacteria”

Section 2

153-155

“Most interestingly, there is also evidence that RNA viruses could be sampled for spacers, which are further reverse transcribed by a Cas1- reverse transcriptase (RT) fusion enzyme (sometimes harboring an additional Cas6 domain), and are subsequently integrated as DNA spacers into the CRISPR array [47–50]” – this statement is somewhat misleading given that to date all but one spacers matches of these arrays actually are to DNA viruses and not RNA viruses. I suggest to change to “that mRNA of DNA  viruses and the genomes of RNA viruses… “

“vicinal to type III” – should be “proximal to type III”

“in representatives of the methanogenic archaeal genus Methanosarcina as well as in

Methanomethylovorans hollandica” – citations are missing here.

249 “even in an attomolar range” – rephrase

Section 3

345 “accomplished degradation of the respective” – rephrase

366 “vicinal” should be “proximal”

Section 4

“in Table 1 (see Tab. 1)” – should be simply “in Table 1.”

“Pyrococcus” – should be in italics

"mechanistical and structural" should be “mechanistic and structural”

“malignant spacers” – should be “deleterious spacers”

Section 5

“organic food sources” – should be “organic carbon sources”

“harsh environments ruling on an early Earth” – should be “harsh environments on an early Earth”

“Furthermore, we could not identify a cas6 in the S. azoricus FC6 genome, indicating that this

organism might not be able to process pre-crRNAs” – does this genome have self-targeting spacers that could maybe explain the inactivation observed?

“as both arrays encompass more than 200 spacers each, thereby also representing the by far longest

CRISPR arrays amongst the Sulfolobales” – given that this will make it difficult for the leader-distal spacers to be expressed, this is an intriguing finding. Is there a region that could serve as an internal promoter, based on its sequence?

I think that order level is not italicized (i.e. Sulfolobales should not be in italics)

“The faith of the cell would depend” should be “The fate of the cell would depend”

“the type III complex could function as a “backup” system to confer immunity” – here it should be clarified if this will require a match to a type III-specific crRNA from a type III array or if there will be compatibility so that the type III can use a type I crRNA if type I is disabled by an ACR. The figure seems to suggest the latter, but this should be made clear in the text.
